# What Differences Exist in Professional Ice Hockey Performance Using Virtual Reality (VR) Technology between Professional Hockey Players and Freestyle Wrestlers? (a Pilot Study)

**DOI:** 10.3390/sports10080116

**Published:** 2022-07-29

**Authors:** Irina Polikanova, Anastasia Yakushina, Sergey Leonov, Anna Kruchinina, Victor Chertopolokhov, Liudmila Liutsko

**Affiliations:** 1Department of Psychology, Lomonosov Moscow State University, 125009 Moscow, Russia; anastasia.ya.au@yandex.ru; 2Department of Biology and Biotechnology, Higher School of Economics (HSE University), 117418 Moscow, Russia; 3Department of Mechanics and Mathematics, Lomonosov Moscow State University, 119234 Moscow, Russia; anna.kruchinina@math.msu.ru (A.K.); psvr.msu@gmail.com (V.C.)

**Keywords:** ice hockey training, wrestlers, transition in sport, virtual reality in sport, motor reaction, puck, steam VR, stance analysis, quantitative analysis, qualitative analysis

## Abstract

There is little research on the study of specific characteristics that contribute to the faster adaptation of athletes during the transition from one sport to another. We used virtual reality (VR) to study the differences between professional ice hockey players and other sport professionals (freestyle wrestlers), who were novices in hockey in terms of motor responses and efficiency performance, on different levels of difficulty. In the VR environment, four levels of difficulty (four blocks) were simulated, depended on the speed of the puck and the distance to it (Bl1—60–80 km/h and 18 m; Bl2—60–100 km/h, distances 12 and 18 m; Bl3—speeds up to 170 km/h and 6, 12, and 18 m; Bl4—the pucks are presented in a series of two (in sequence with a 1 s interval)). The results of the study showed that the hockey professionals proved to have more stable movement patterns of the knee and hip joints. They also made fewer head movements as a response to stimuli during all runs (0.66 vs. 1.25, *p* = 0.043). Thus, working out on these parameters can contribute to the faster adaptation of wrestlers in developing professional ice hockey skills.

## 1. Introduction

### 1.1. Use of VR Technologies in Sports

Over the last decade, virtual reality (VR) technology has been actively used to develop, train, and improve sports skills [1,2,3,4,5]. Simulating sports situations in VR is becoming increasingly popular for the purpose of assessing skill formation and learning. This is related to researchers having control over positional and temporal task parameters and the number of repetitions without changing environmental conditions, while also having the ability to choose the quality and quantity of feedback [6,7].

Nevertheless, there are some limitations to the use of VR in sport training. Some studies demonstrate the inconsistency of virtual reality technologies as a method of professional skills formation. These studies point out that virtual reality does not always contribute to the transfer of the trained skill to reality [8,9]. In addition, the virtual environment does not always have a full presence effect [10,11]; for example, in the VR environment, it is quite difficult to organize the training of skills related to contact with other athletes or team player skills [12]. 

However, VR allows the athlete to train cognitive functions as well as skill subcomponents [13] necessary for high performance (e.g., attention distribution, long-term memory, etc.); it also allows them to train proprioceptive or automatic movements that work quicker due to multiple repetition; these can be tailored to the athletes specific needs with the integration of vision and body-centered (proprioception) systems of coordinates [14,15,16], allowing them to practice movements without risks to the athlete’s health, because the risk of injury is reduced. Additionally, VR allows the athlete to train beyond their abilities in a real-world environment, making it easier to effectively track the athlete’s progress [1,17,18].

In the Ijsselsteijn’s study using exercise bikes, it was shown that the feeling of immersion created by the virtual computer environment has a positive effect on the motivation of athletes and increases the level of satisfaction with the sport [19]. Bideau and colleagues in their research observed that goalkeepers in handball react equally to both real and virtual attacks [20]. Pavla Satrapová, Tomáš Perič and Adam Rulík in their study on 197 hockey players showed that virtual reality can be a good tool for specific hockey training, especially to measure specific hockey thinking, including both cognitive functions and specific game skills [21]. Thus, we can see that using virtual reality, in spite of some limitations and difficulties, has great potential, including in high-performance sports.

### 1.2. Distinctive Features of Hockey Training 

Hockey players’ success is conditioned by specific skills associated with high muscular and cognitive exertion, as well as the intensity of working at maximum performance for a certain period of time [19,20,21,22,23,24]. A hockey player’s performance depends, as well, on skating skills, which in turn require optimal coordination of lower body joints and muscular strength for both body movement and dynamic stability [25,26,27,28]. That is why the most characteristic indicators of hockey players’ skills are those related to skating speed, reaction speed, and movement coordination. These indicators allow us to distinguish professional players from novice players [29,30,31,32]. 

In the context of the *mechanical outcomes* (e.g., kinematics, technique, hip movement, knee flexion), the following examples can be mentioned. A study by Blanár and colleagues [33] showed that the better an athlete’s lower extremity explosive strength and balance skills developed, the more successful he or she was in the exercise. In addition, hockey players’ skating skills are analyzed through kinematic analysis of changes in hip, knee, and ankle joint angles [34,35]. A study by Shell and colleagues [36] demonstrated that changes in hip and knee joint angles and their kinematics can significantly affect the speed of hockey players, with male hockey players having a significantly higher speed compared to females [36]. A study by Upjohn and colleagues [37] also revealed differences in the angles of the knee and hip in hockey players with different skill experience. The authors noted that more experienced and titled hockey players have greater hip and knee deflection angles during skating compared to less experienced ones.

Additionally, a study by Lafontaine [38] demonstrated that changes in knee joint angles are significantly related to hockey players’ skating speeds; that is, the greater the speed an athlete gains to perform a maneuver, the more significant are the changes occurring in knee joint angles [38]. 

A study by Bracko [39] found that elite hockey players’ (N = 23) acceleration during game skating (dynamic acceleration) differed significantly from the dynamic acceleration of novices. At the same time, the skating speed of novices and professionals at the start does not differ. Thus, it can be assumed that not only skating speed, but also stability indices and susceptibility to game movements can be considered specifically to distinguish the skill level of hockey players.

In the context of *perceptive-cognitive aspects*, it should be noted that the skill of a hockey player is determined by the ability to anticipate, to switch quickly between tasks, and to be resistant to increasing stimuli around him, while performing the leading activity with increasing surrounding stimuli [40,41,42]. Many studies confirm that expert-level athletes outperform novices on sport-specific tasks that include decision making, prediction, spatial memory, response accuracy and response time [43,44,45,46,47]. 

Appelbaum and Erickson point out that training athletes on demanding visual, cognitive, or oculomotor tasks can improve their ability to process and respond to what they see, thereby improving athletic performance [48]. Veronique and colleagues’ study showed, using the Canadian Women’s National Team preparation for the 2020 Olympics, the goalkeepers’ importance and the validity of using VR training to develop perceptual-cognitive skills (i.e., anticipatory training using video occlusion, eye-hand coordination, and visuomotor drills) [49]. A. Romeas and colleagues’ study of soccer players observed the effectiveness of transferring the perceptual-cognitive skill of tracking three-dimensional objects, necessary for high performance in soccer, from the laboratory to the soccer field [50]. This in turn also proves the adequacy of using cognitive training to improve sports skills. 

Analyzing the literature, we did not find studies that compared the kinematic characteristics of the hips, knees, and head in professional hockey players and novices. However, it seemed important for our research to focus on these characteristics and to examine possible differences. At the same time, we can point to a number of studies that analyzed the kinematic characteristics of athletes to identify the most optimal parameters and for other purposes. In the study of Zecha and colleagues, it was shown that the extraction of kinematic parameters of athletes allows one to obtain feedback on the effectiveness of the training process as well as to monitor progress in mastering the mastership [51,52]. Lee and Kim’s study indicated that the 4-week virtual reality (VR) sports workout program developed in this study proved to be suitable as a VR sports workout to improve body composition and health [53]. A study by Rao and colleagues observed the effectiveness of using kinematic analysis on the movements of professional marksmen [54]. The authors suggest that the results should be effectively used in the training of novice athletes. Bacos and Carroll suggest using 3D immersive virtual environments to analyze body movements during e-learning interactions [55]. Zarzycki and colleagues showed significant correlations with the psychological readiness of athletes to return to training based on kinematic analysis after leg surgery [56]. 

### 1.3. Using VR in Hockey

For VR ice hockey players training, it is relevant to create a specialized environment using a virtual reality system to analyze their professional skills; this can be also used both for training and the observation of dynamic changes. In general, there are quite a few studies on the use of VR in hockey. For example, in a study by Buns [57], based on the idea that virtual training is an effective tool to improve sportsmanship, it was shown that practicing hockey elements in a virtual environment affects the effectiveness of training in the real world. Compared to the control group, participants in the experimental group significantly improved the accuracy and speed of hockey shots on the ice [58]. Additionally, a study by Tyreman and colleagues [59] demonstrated that in a simulated three-dimensional environment, the reaction time of professional hockey goalkeepers and novices did not differ. Nevertheless, a special type of shot, characteristic only of professionals, was identified. That is, it was the detailed comparative analysis of the indicators obtained using the three-dimensional space that allowed us to determine the differences between professional goalkeepers and novices [11]. In general, Tyreman suggests that three-dimensional VR can be useful in improving reaction times, anticipation times, or strategies [50]. Katz and Tyreman with colleagues suggest that virtual environments, despite all their complexity and cost, have incredible potential to change the way coaches and athletes approach training and results. The future development of VR systems will include many aspects of monitoring, management and coaching [11].

Studies comparing the specific characteristics of hockey players and other athletes are scarce. At the same time there are unofficial reports that many hockey players train in martial arts skills [60,61,62]. Many people think that they learn how to fight, but in fact they train skills such as anticipation, weight balance, body positioning, how to use some different body parts and mechanics, so they can become better athletes overall, which helps them to achieve great success in hockey [60]. Additionally, martial arts help to improve reaction speed and balance and helps to train power and endurance, which might be needed on the ice [63]. Hockey also requires a very high level of concentration as a very high reaction speed and quick decision making are essential and can cost the team to win. Due to the fact that hockey players and martial artists do possess some necessary skills—anticipation, hand and eye coordination, reaction speed, strength and endurance—many professional hockey players purposefully train in martial arts skills [64]. There are several studies that show evidence that skills obtained in some sports can contribute to improved performance in other sports. The article by Soberlak and Côté, for example, shows that the skills athletes learn in different sports when they are young can have a positive impact on their skill level in professional hockey [65]. Likewise, Hodge and Deakin come to similar conclusions about the usefulness of the contribution of various sporting skills to the development of skill levels in a particular sport [66].

Therefore, in order to determine the specific features of professional hockey players, we decided to compare the features of performing technical exercises in professional hockey players and wrestlers (all novices in hockey). This is due to the fact that wrestling and hockey have some similar features: both hockey players and wrestlers have well-developed skills in anticipating the actions of their opponent and their reaction speeds to reflect external actions [46,67,68]. At the same time, the significant differences between these sports will allow us to highlight specific characteristics directly for hockey players. Thus, the aim of the present study was to determine whether there were any differences in performance between professional hockey players and freestyle wrestlers in the task of shooting pucks at different levels of difficulty, using virtual reality (VR) technology between professional hockey players and freestyle wrestlers (novices in hockey).

The main goals of the study were: (1) to describe the main specific features that determine the performance efficiency of professional hockey players and wrestlers based on motor parameters and the success of puck kicking (analysis of changes in knee and hip joint angles; head movements; stick movement; speed of response to puck presentation; the number of hit and missed pucks); (2) to design a VR environment simulating a hockey field with the ability to present pucks of different difficulty levels, as well as the ability to register the subjects’ motor responses and the success of puck kicking; (3) perform a series of experiments and record, using VR, motor reactions and performance in the task of hitting pucks at different levels of difficulty, both in a group of hockey players and a group of wrestlers; (4) to compare groups of hockey players and wrestlers in professional ice hockey performance; (5) to determine similar motor performance characteristics for hockey players and wrestlers; and (6) to determine hockey player-specific motor performance characteristics.

### 1.4. Hypotheses

Based on the analysis and the stated aim of the study, we propose the following research hypotheses:
The visual qualitative analysis of changes in knee and hip joint angles in the hockey group and the freestyle wrestling group will be qualitatively different in Blocks 1–4.Significant differences in the number of pucks taken in Blocks 3 and 4 will be determined between the freestyle wrestling group and the hockey group. We expect that hockey players have developed specific anticipation skills that allow them to hit pucks at high speeds and maneuver their stick when presented with two pucks.Significant differences in reaction rate in Blocks 3 and 4 will be determined between the freestyle wrestling group and the hockey group.Significant differences in posture, head movements, and in the angles in knee and hip joints during puck kicking in all blocks will be determined between the freestyle wrestling group and the hockey group.The hockey group will be characterized by higher results both in the number of pucks scored mainly in Blocks 3 and 4 (the most difficult tasks) and in support of postural balance during the whole experiment, due to the formation of professionally important qualities necessary for a hockey player.The freestyle wrestling group will show (a) high performance in Block 1 (easy level of difficulty) and Block 2 (medium level of difficulty) due to the formation of the skill of anticipation, which is a professionally important quality in martial arts; and (b) low performance in supporting postural balance throughout the experiment due to the lack of formation of a professionally important skill—the hockey stance.

## 2. Materials and Methods

### 2.1. Participants 

The study involved 20 participants, 13 of them being professional ice hockey players (age = 20 ± 2.5 years; mean duration of training experience 14.18 ± 3.8 years) and seven being freestyle wrestlers (age = 19 ± 1.9 years; mean duration of training experience 8 ± 6.10) who were novices in hockey. All of them participated on a voluntary basis with previously signed consent, and with prior approval by the Ethical Committee of the Russian Psychological Society (March 2021), in accordance with the Helsinki Declaration [69]. Inclusion criteria for the sample were: age over 18 years, ability to stand on skates, presence of normal or corrected vision. Exclusion criteria were women due to the need to make adjustments for the menstrual cycle.

### 2.2. Description of the Virtual Environment and the Subjects’ Tasks

Virtual reality (VR) using the SteamVR Tracking 2.0 system was used to study the differences between professional ice hockey players and novices in hockey with another sports background (freestyle wrestlers) in terms of motor responses on different levels of difficulty. VR displays the environment from the point of view of a hockey player standing on the goal line (Figure 1 and Figure 2). The subject views the body of his virtual avatar in the first-person and holds a stick (the stick in the virtual environment is the same as in reality). There are no other players on the field. The player’s equipment corresponds to that of a team player (except goalkeeper), since the main goal of the study is to train and improve the individual motor skills of team players in ice hockey (except goalkeeper). In addition, the equipment of goalkeepers, as well as their motor patterns and game tactics, are significantly different from team players. 

In the VR environment, people were completely absent—both spectators and other players. We allocated a unit of analysis of a hockey play—a hockey player who hits the puck. At the same time, we modeled different levels of difficulty for the selected unit of analysis. Thus, the VR environment we used does not simulate the game entirely but simulates a specific game unit—a hockey player who hits the puck. 

Pucks were presented in random order with different speeds (5 speeds: 60, 80, 100, 130, 170 km/h), from different distances (18, 12, and 6 m), different locations (right/left/center), and different heights (at ice level, and at a height of about 50 cm).

The choice of speeds and distances was made in several stages. The preliminary idea was to make a wider range of values, in increments of 10 km/h. However, preliminary measurements showed that presenting different speeds in small increments is subjectively almost the same, but greatly lengthens the experiment. To determine the minimum and at the same time a sufficient number of parameters (different speeds and distances), we conducted test recordings, including the participation of experts and a hockey coach. For the choice of speeds, we focused on the widest possible range: from slow (when the hockey player sees the puck accurately), for example 60 km/h; medium speeds—80, 100 and 130 km/h; and the highest possible speed—170 km/h.

All subjects before the experiment were given detailed instructions describing all tasks, blocks, and levels of difficulty; they were given information about which pucks counted—only the puck’s hook hit; and subjects (not hockey players) were shown in detail what a hockey posture was and how to maneuver their posture correctly. There was no additional practice before Block 1.

The player was instructed as follows: hit all the pucks flying into the net with a stick. Pucks are presented randomly from various distances and directions and with different initial speeds. Each puck starts with a specific click sound. Just before the puck is presented, the rink section from which the puck starts, is illuminated with yellow light (Figure 3). Pucks that are not hit with a stick do not count. The simulation of pucks shooting includes three blocks of difficulty depending on the speed of the pucks and their distance from the subject. The closer the distance and the higher the speed, the more difficult the block is. Block 1 is the simplest, with low puck speeds (60–80 km/h) and long distances to the puck (18 m), so even beginners with no hockey experience would manage to hit it. Block 2 is more difficult, speeds are higher (60–80 km/h, and 100 km/h), average distances to the puck are longer (12 and 18 m). Block 3 is for medium skill level; it is a difficult mode with high speeds (up to 170 km/h) and all distances, including close ones (6, 12, and 18 m). Block 4 is the most difficult one, where the pucks are presented in a series of two (in sequence with a 1 s interval). Speeds of pucks include all ranges (60 to 170 km/h) and all distances (6, 12, and 18 m).). Block 5 is where the subject is not supposed to hit the pucks, but only carefully watch and track the pucks, which are presented with random difficulty. The interval between the pucks is 3 s with a click sound. After each puck, the player looks to the center. The results of Block 5 were not considered in this article. These data require a separate analysis of oculomotor characteristics and strategies. We plan to examine them separately and present them in a separate publication, in which we will certainly compare them with the data already presented. In Blocks 1, 2, 3 and 5 were presented 14 pucks each. In Block 4 were presented 28 pucks, respectively, since the task was to knock two pucks at once.

### 2.3. Analyzed Parameters

The movements of the subject’s posture and the stick were registered using the SteamVR Tracking 2.0 system: (1)Visual qualitative analysis of changes in knee and hip joint angles;(2)Changes in the angle of the knee joint (right and left knee);(3)Angle changes at the hip joint (right and left side);(4)Head movements;(5)Stick movement (average speed per measurement cycle);(6)Speed of response to puck presentation;(7)The number of hit and missed pucks.

Trackers were attached to the shin guards, hips, chest, gloves, and stick (Figure 4). The changes in the angles in knee and hip joints were chosen to be analyzed. To fully evaluate the stance and mistakes, we considered the relative position of trackers on the right and left hips, alongside the tracker of the head position (tracked by VR helmet) in relation to the legs. 

These values were calculated as follows: the time window was 0.1 s, the moments of motor reaction were selected as peaks with a greater height than the tripled value of the mean-square deviation of the corresponding value during a still standing (30 s before stimuli).

Visual qualitative analysis of changes in knee and hip joint angles was carried out for all subjects, but for the pictures, examples of recordings in a professional hockey player and a beginner were chosen.

### 2.4. Data Analysis

The study used a mix methodology of quantitative and qualitative data analysis. Descriptive and comparative analysis was performed with the use of IBM SPSS Statistics 22 for Windows. We used a nonparametric Mann–Whitney criterion (α = 0.05) to compare a group of hockey players and a group of freestyle wrestlers (not hockey players). 

Visual qualitative analysis of changes in knee and hip joint angles was also performed to identify visual specific motor characteristics.

For the statistical analysis, the following parameters were calculated:Hitting pucks (%);Motor response time to the puck’s arrival (RT1);Stick response time (RT2);Stance analysis (the changes in the angles in knee and hip joints, head movements).

RT1 was counted as the time from the moment of the puck’s departure (this moment was synchronized and registered using the SteamVR Tracking 2.0 system) to the first motor response exceeding three-sigma based on the knee and hip joint angles analysis. 

RT2 was counted from the moment of the flight of the puck (this moment was synchronized and registered using the SteamVR Tracking 2.0 system) to the moment of “collision” with the hook of the stick (in a virtual environment this is the moment of intersection of the puck trajectory and the hook of the virtual stick). In cases where the puck was missed, the flight time of the puck was counted from the moment the puck flew out to the moment it crossed the virtual goal line.

## 3. Results

### 3.1. Visual Qualitative Analysis 

Figure 5, Figure 6, Figure 7 and Figure 8 show a graphical representation of the dynamics of the standard deviation of changes in the angles in the knee and hip joints during puck shooting in hockey players and wrestlers in the VR environment. Based on the visual analysis, we can make a preliminary report about the characteristics of posture. The movements of professional hockey players are more stable and precise (Figure 5 and Figure 6) than those of wrestlers (Figure 7 and Figure 8). This is reflected in the symmetrical and synchronous changes in knee and hip joint angles. The quantitative values of these parameters are presented and considered below.

### 3.2. Value of Hitting Pucks

The analysis revealed no significant differences between the groups of hockey players and wrestlers in the number of pucks hit in the four blocks (Table 1).

### 3.3. Response Time

The results showed significant differences in the stick-to-puck response time in Block 1 (with low puck speeds (60–80 km/h) and long distances to the puck RT2) (1.13 vs. 1.58, *p* = 0.036), and Block 3 (with high speeds (up to 170 km/h) and all distances RT2) (1.08 vs. 1.66, *p* = 0.021) (Table 2). No significant differences in motor responses to the introduction of stimulus (RT1) were revealed; this means that the motor response to the stimulus was approximately the same in the group of hockey players and freestyle wrestlers.

### 3.4. Stance Analysis

The stance analysis revealed significant differences between the hockey players and the wrestlers in maintaining the head’s position (*p* < 0.05), showing that average head movement indicators of professional hockey players are lower than those of novices (Table 3).

## 4. Discussion

In this study, we performed a comprehensive investigation to understand the differences that exist in professional ice hockey performance using virtual reality (VR) technology between professional hockey players and freestyle wrestlers.

We developed a virtual environment simulating a hockey field and simulating the training of hockey skills. We allocated a unit of analyses of a hockey figure—a hockey player who hits the puck. Several tasks with different levels of complexity were modeled in the VR. In the VR, we recorded motor reactions and performance in both the hockey and wrestler groups in the task of hitting pucks with different complexities and compared the two groups according to the hypotheses.

As a result of the study, Hypothesis 1—the visual analysis of changes in knee and hip joint angles in hockey group and the freestyle wrestling group will be qualitatively different—was confirmed. We showed that, according to the results of the qualitative visual analysis, the characteristics of posture in the group of hockey players are more stable and precise, and the changes in knee and hip joint angles are more symmetrical and synchronous. This indicates the greater optimization of resources for the performance of a motor skill in hockey players, as well as the better development of visual-spatial skills.

Hypothesis 2—the hockey players and the freestyle wrestlers will not differ in the number of pucks hit in Blocks 1 and 2 but will differ in the number of pucks hit in Blocks 3 and 4—was not confirmed. The results showed no differences in both the number of pucks blocked and the number of pucks hit in all series. In our opinion, this may be due to the fact that for hockey players and wrestlers, such skills as reaction speed and anticipation are equally important, which helps them in puck repulsion. In wrestling, it is also necessary to be able to master spatial situations well enough and to react quickly to the actions of the opponent [70].

Hypothesis 3—the hockey group and the freestyle wrestling group will not differ in reaction rate in Blocks 1 and 2 but will differ in reaction rate in Blocks 3 and 4—was partially confirmed. Thus, it was shown that there are significant differences in the stick reaction rate (RT2) to the presented pucks in the first block (the lowest presentation rate) and the third block (the highest presentation rate). The differences in hockey stick reactions in the first block may be related to the fact that wrestlers needed more time to adapt to the task as opposed to hockey players, for whom it was similar to their usual activity. Similar results are consistent with similar studies confirming faster adaptation to the VR environment among those who actually experienced similar activities [4,58,71]. It is worth noting that no differences were found in the stick response to pucks presented (RT2) in a block with an almost simultaneous presentation of two pucks (Block 4). In our opinion, this may be due to the fact that the tasks of this block in the VR environment may have been performed worse by all participants due to the appearance of a sense of fatigue in virtual reality [9,10]. At the same time, no significant differences in motor responses to stimulus appearance (RT1) were found; that is, the motor response to stimulus appearance in both the hockey and wrestling groups occurred approximately in the same way. The results obtained testify to the good formation of spatio-temporal abilities, both for the group of hockey players and the group of wrestlers.

Hypothesis 4—hockey players and the freestyle wrestling group will differ in posture, head movements, and in the angles in knee and hip joints during puck shooting in all blocks—was confirmed. Professional hockey players showed a significantly smaller amplitude of head oscillation throughout the study, which may indicate a well-formed spatial factor: that is, they understand their body position quite well during certain actions, they do not need to observe the full trajectory of puck movements, and they notice them faster [15,72,73,74,75]. This may indicate that indicators of stability of movement of certain body parts, in particular the head, can be a criterion indicating the high skill of a hockey player, not only reaction speed.

Hypothesis 5—the Hockey group will be characterized by higher results both in the number of scored pucks mainly in blocks 3 and 4 (the most difficult tasks), and in maintaining postural balance throughout the experiment, due to the formation of professionally important qualities necessary for hockey players—was partially confirmed. The results showed that hockey players and freestyle wrestlers do not differ significantly in the number of pucks taken, but they differ in positional characteristics, in particular in the smaller amplitude of head oscillation during the passage of the experiment. This may indicate the formation of spatial anticipation in hockey players, which helps them to anticipate the trajectory of the puck flight without visual monitoring.

Hypothesis 6—the Freestyle Wrestling Group would show (a) high scores in Block 1 (easy level of difficulty) and Block 2 (medium level of difficulty) due to the formation of the anticipation skill, which is a professionally important quality in martial arts; and (b) low scores in maintaining postural balance throughout the experiment due to the unformed professionally important skill—the hockey stance—was confirmed. The group of wrestlers showed high performance in puck rejection, apparently due to already formed anticipation, necessary in martial arts. However, due to the lack of postural automatisms specific to hockey players, wrestlers showed low efficiency in maintaining postural balance.

Thus, our study is original in terms of approach as we do not simply compare novices to professionals, we compare athletes from different sports that require specific professionally important abilities needed for high performance. We used VR to study the differences between professional ice hockey players and other sport professionals (freestyle wrestlers) who were novices in hockey in terms of motor responses and efficiency performance on different levels of difficulty. We found metrics specific to high performance in hockey. Thus, we have concluded that the presence of these characteristics can contribute to a faster adaptation in ice hockey. Additionally, our results showed that virtual reality could be successfully used to practice and to train specific skills necessary to achieve high performance in hockey.

### Limitations

This study has some limitations. The first is the small sample size. Since the study involved hockey players with a high level of sportsmanship (M = 14.2 ± 2.3 years of average hockey experience, range: 8–20 years old), access to whom is rather limited—especially given the coronavirus pandemic—the sample was not large in total. Moreover, each test recording took a rather long period of time due to the need of putting on hockey equipment, fixing the sensors, and adjusting the VR equipment. It should also be noted that only two groups had an impact on the results. Some experts suggest that a small sample size is not necessarily a major limitation as the available cohorts for both professional ice hockey players and professional wrestlers (and in particular their availability for research) are limited due to expertise levels (Müller, Brenton and Rosalie (2015)). At the same time, we intend to expand the sample, including a comparison of the obtained data on a reference group, such as non-athletes.

Another important limitation of our study is the use of VR techniques to diagnose the professional skill of hockey players. VR is still a method that does not always show stable skill transfer results. Therefore, other indicators (cognitive tests, on-field exercises, etc.) should also be considered to assess the professional skill of hockey players. Further studies with an increased sample size and the addition of other sport types, or people without any professional sport training, might help to better understand the differences in acquiring hockey professional skills by use of VR. Longitudinal studies would be beneficial to track the dynamics and speed of acquiring professional skills with the use of VR.

As part of this study, we obtained results reflecting the differences between professional ice hockey players and freestyle wrestlers in performing the task of hitting pucks at different levels of difficulty. Since it was a pilot study, and due to the difficulty of accessing professional athletes, including COVID-19 limitations, the sample size was not huge (though enough that the coherent statistical power of 0,80 was reached), therefore these results are difficult to generalize to a large sample. More studies, including comparisons with other types of sports, could be analyzed for common bases (similarities) and those parameters that differ and need to be determined to facilitate a more fluent transition from one type of sport to another.

## 5. Conclusions

In our study, it was shown, using virtual reality (VR) technology, that there are differences between professional ice hockey players and freestyle wrestlers in performing the task of hitting pucks at different levels of difficulty; these differences are reflected in hockey stance and stick response time. At the same time, the group of wrestlers who were novices in hockey but who had another sports background, did not differ from the group of professional hockey players in the number of hit and missed pucks, as well as the motor response time to the puck’s arrival.

This shows that when adapting from one sport to another (not even similar sports), some athletes can adapt much more quickly to the demands of the new sport (in our case, hockey) if they have certain important qualities already developed. In our case, this quality is reaction time.

We also showed that virtual reality can be successfully used to practice and train specific individual parameters necessary for high performance in hockey, such as motor response time to the puck’s arrival, stick response time, puck catching, and stance analysis (head movements, and changes in the angles in knee and hip joints).

## Figures and Tables

**Figure 1 sports-10-00116-f001:**
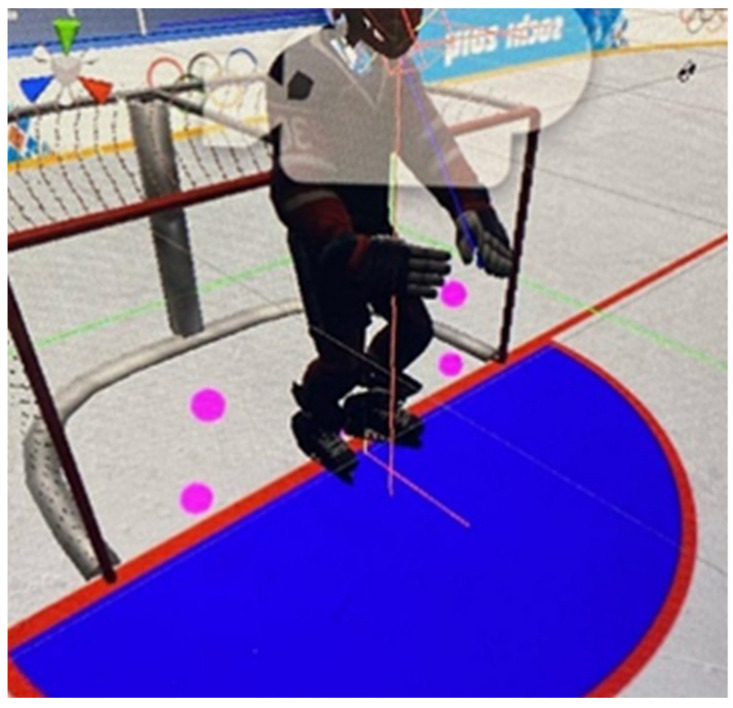
Avatar.

**Figure 2 sports-10-00116-f002:**
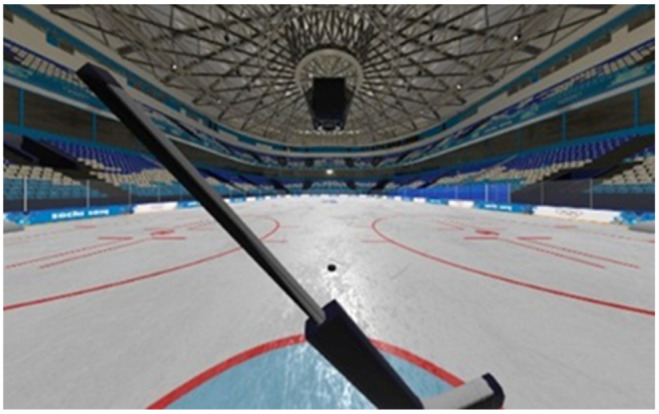
Environment visualization in VR helmet.

**Figure 3 sports-10-00116-f003:**
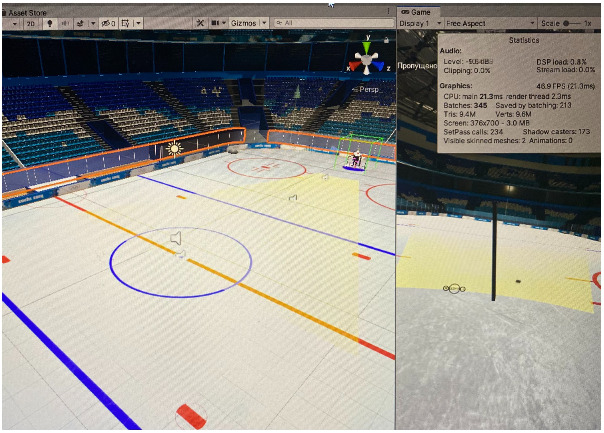
Rink section illumination (general view on the left, view in front of the subject on the right).

**Figure 4 sports-10-00116-f004:**
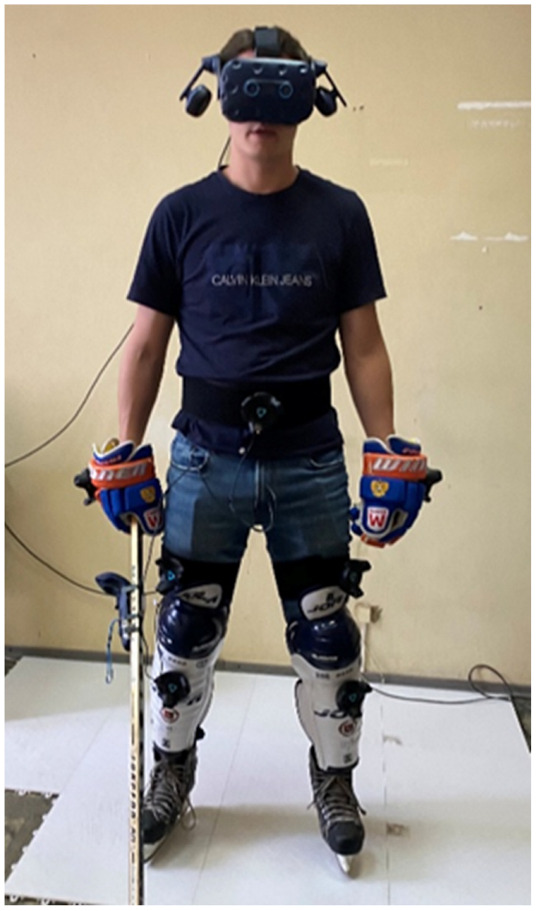
Subject with the trackers attached to his shin guards, hips, chest, gloves and stick.

**Figure 5 sports-10-00116-f005:**
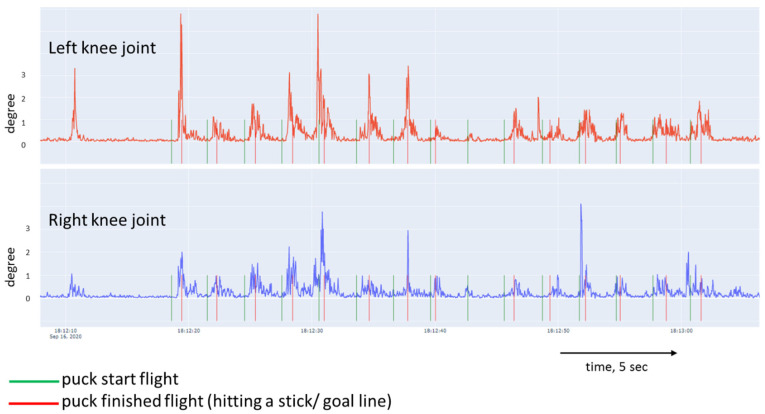
Change in the angle of the knee joint of professional ice hockey player. (19 years of training experience) in Block 1.

**Figure 6 sports-10-00116-f006:**
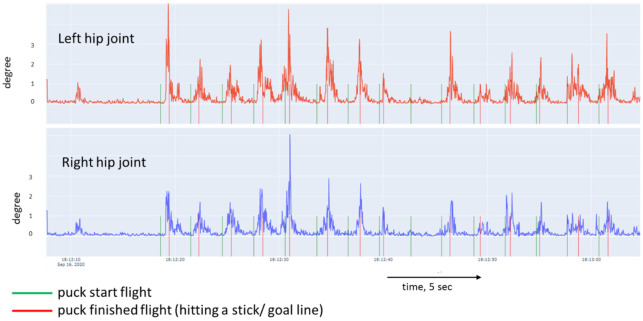
Change in the angle of the hip joint of professional ice hockey player. (19 years of training experience) in Block 1.

**Figure 7 sports-10-00116-f007:**
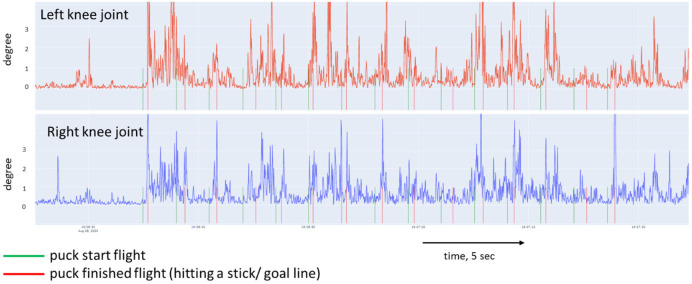
Change in the angle of the knee joint of freestyle wrestler in Block 1.

**Figure 8 sports-10-00116-f008:**
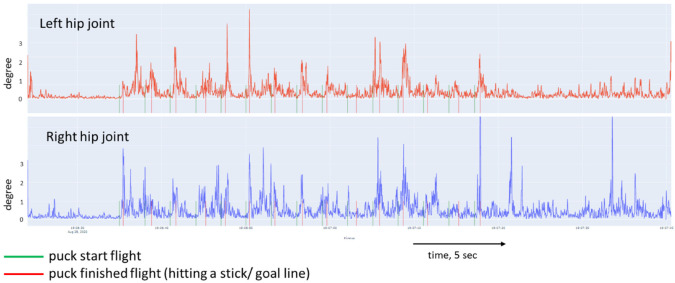
Change in the angle of the hip joint of freestyle wrestler in Block 1.

**Table 1 sports-10-00116-t001:** Values of hitting and missed pucks, data are presented as a mean and +/− standard deviation, sec.

Group		Hockey	Freestyle Wrestlers	U M-W Criterion	*p*-Value	Cohen’s d
Hitting pucks	block 1	7.77 ± 2.9	6.63 ± 3.2	42.50	0.809	0.40
	block 2	5.85 ± 1.7	5.56 ± 2.6	28.50	0.170	0.13
	block 3	4.85 ± 1.7	3.44 ± 1.9	27.00	0.312	0.78
	block 4	8.00 ± 2.3	7.44 ± 1.8	36.00	0.790	0.27

**Table 2 sports-10-00116-t002:** Average values of motor response time (RT1) and stick response time (RT2), sec (M ± SD), sec.

Group		Hockey	Freestyle Wrestlers	U M-W Criterion	*p*-Value	Cohen’s d
RT1	block 1	0.44 ± 0.15	0.38 ± 0.19	39.00	0.800	0.35
	block 2	0.41 ± 0.14	0.33 ± 0.12	39.00	0.606	0.62
	block 3	0.37 ± 0.17	0.31 ± 0.08	24.00	0.692	0.45
	block 4	0.29 ± 0.07	0.29 ± 0.06	31.00	0.251	0.00
RT2	block 1	1.13 ± 0.35	1.58 ± 0.73	19.00	0.036 *	0.79
	block 2	0.96 ± 0.42	1.55 ± 0.92	35.00	0.405	0.83
	block 3	1.08 ± 0.65	1.66 ± 1.54	10.00	0.021 *	0.49
	block 4	0.98 ± 0.30	1.50 ± 0.71	28.00	0.166	0.96

* level of statistical significance *p* ≤ 0.05. Legend: RT1, motor response time; RT2, stick response time.

**Table 3 sports-10-00116-t003:** Average values of motor response time of different body parts, degree ± SD.

Group	Hockey	Freestyle Wrestlers	U M-W Criterion	*p*-Value	Cohen’s d
Knee, right	0.81 ± 0.26	0.75 ± 0.27	25.00	0.151	0.25
Knee, left	0.77 ± 0.26	0.72 ± 0.22	38.00	0.735	0.21
Hip, right	1.01 ± 0.34	1.13 ± 0.27	38.00	0.735	0.37
Hip, left	1.13 ± 0.34	1.13 ± 0.22	41.00	0.933	0.02
Head	0.66 ± 0.85	1.26 ± 0.86	18.00	0.043 *	0.70

* level of statistical significance *p* ≤ 0.05.

## Data Availability

Data available at GitHub: https://github.com/IPolikanova/VR_hockey (accessed on 7 July 2022).

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
