# Peer review of "What Differences Exist in Professional Ice Hockey Performance Using Virtual Reality (VR) Technology between Professional Hockey Players and Freestyle Wrestlers? (a Pilot Study)"

_sports, 2022, doi:10.3390/sports10080116_

Round 1
Reviewer 1 Report
The article seems to be interesting from the cognitive point of view, and after some improvements, it will become a scientific article.
First of all, you should consider choosing the right keywords, thanks to which it will be possible to extend the scope of the article indexation (you should not use the same keywords as in the article title).
The abstract should explain the measurement methodology in more detail, please describe each of the blocks. Also, please provide more details about the participants; by the way what is "Mage"? My guess is the average age - please change to: (age: 20.0 + -2.5 years). In lines 21-23, please provide units for the presented values.
On line 112, please correct "ice2" to "ice". Please also consider the following works in the introduction: doi: 10.1515 / hukin-2015-0165; doi: 10.2478 / hukin-2014-0126.
The research objective in lines 124-128 is general, followed by specific objectives in the following lines 129-135; unfortunately, no details were given about what will be tested.
The concept of "Blocks 1-3" appears in the hypotheses, but nowhere have I found what it means - it should be clarified.
Participants
Please describe the respondents as accurately as possible; age, height and weight, criteria for inclusion and exclusion in the research, training period, etc. Did the respondents have an opportunity to get acquainted with VR?
Data analysis
Please specify how the points were calculated, explain why it was decided to use non-parametric statistics (Mann-Whitney test), how the effect size was calculated and what criteria were adopted for estimating the size.
Results
Is it possible to present 5,6,7,8 Y axis (ordinate) in the same scale in the graphics? Unfortunately, each of the graphs has a different scale, which makes comparison difficult.
Table 1: the title of the table could be: Values of hitting and missed pucks, data are presented as a mean and + - standard deviation "; please provide units for the" Hockey "and" Freestyle wrestlers "results; I think you can give it in the table header.
Table 2: please provide units for the reported results "Hockey" and "Freestyle wrestlers"
Author Response
Thank you for the good review of our manuscript!
We tried to answer all of your comments.
Comments and Suggestions for Authors:
The article seems to be interesting from the cognitive point of view, and after some improvements, it will become a scientific article.
Answer: Thank you very much!
First of all, you should consider choosing the right keywords, thanks to which it will be possible to extend the scope of the article indexation (you should not use the same keywords as in the article title).
Answer: We added more relevant keywords and eliminated less relevant
The abstract should explain the measurement methodology in more detail, please describe each of the blocks. Also, please provide more details about the participants; by the way what is "Mage"? My guess is the average age - please change to: (age: 20.0 + -2.5 years). In lines 21-23, please provide units for the presented values.
Answer: Mage = MEAN (average) age. We corrected as suggested by the reviewer "(average age=20±2.5 years)"and added units. We also completely reformulated the abstract to reduce the volume to be fit in 200 words.
On line 112, please correct "ice2" to "ice".
Answer: done.
Please also consider the following works in the introduction: doi: 10.1515 / hukin-2015-0165; doi: 10.2478 / hukin-2014-0126.
Answer: done.
The research objective in lines 124-128 is general, followed by specific objectives in the following lines 129-135; unfortunately, no details were given about what will be tested.
Answer: done. We have added a more detailed description of the main goals.
The concept of "Blocks 1-3" appears in the hypotheses, but nowhere have I found what it means - it should be clarified.
Answer: done. We have added a more detailed description of "Blocks 1-3" in abstract.
Participants
Please describe the respondents as accurately as possible; age, height and weight, criteria for inclusion and exclusion in the research, training period, etc. Did the respondents have an opportunity to get acquainted with VR?
Answer:
Detailed information about the study participants (age, mean duration of training experience, all study results) is available at GitHub: https://github.com/IPolikanova/VR_hockey. The link is provided at the end of this publication. Height and weight were not recorded since they were not considered in the current study protocol. But this point will be considered in the future studies.
Also we added the next information to this section: Inclusion criteria for the sample were: age over 18 years, ability to stand on skates, presence of normal or corrected vision. Exclusion criteria were women due to the need to make adjustments for the menstrual cycle, but will be considered inb the future studies.
Before the experiment, participants only had the opportunity to familiarize themselves with the hockey VR environment. Participants had no opportunity to test the system. For this purpose, block 1 was made simple with slow speeds. One of the main factors why we rejected the trial was to optimize study time. In the current experiment, the total test duration was about 8-10 minutes (recording only).
Data analysis
Please specify how the points were calculated, explain why it was decided to use non-parametric statistics (Mann-Whitney test), how the effect size was calculated and what criteria were adopted for estimating the size.
Answer: Nonparametric statistics were chosen due to the small sample of subjects in both groups. Effect size and statistical power were calculated using an online calculator and were 1,06 and 80%correspondently.
The small sample size was due to a number of factors:
- Limited access to professional athletes, limited free time for professional athletes
- The limitations associated with COVID-19
- the duration of the experimental procedure (approximately 40 minutes per subject)
Results
Is it possible to present 5,6,7,8 Y axis (ordinate) in the same scale in the graphics? Unfortunately, each of the graphs has a different scale, which makes comparison difficult.
Answer: done.
Table 1: the title of the table could be: Values of hitting and missed pucks, data are presented as a mean and + - standard deviation "; please provide units for the" Hockey "and" Freestyle wrestlers "results; I think you can give it in the table header.
Answer: done.
Table 2: please provide units for the reported results "Hockey" and "Freestyle wrestlers"
Answer: done.

Reviewer 2 Report
The work presented is interesting, however, it presents a series of difficulties that I believe do not make it suitable for publication:
1. They only use physical variables. It has been shown that approximately 30% of sports performance is due to psychological variables. This is a fact that the authors acknowledge in the introduction.
2. The hypotheses are inadequately stated. Hypotheses should only be formulated on contrast statistics and qualitative analysis. Hypotheses should be formulated as follows: "Significant differences in [variable] will be determined between the control group and the experimental groups".
3. There are no references to the authorization of the ethics committee or to the Declaration of Helsinki.
4. The sample is very small.
5. It is necessary to perform a generalizability analysis to determine the generalizability of the results.
6. Taking into account that it uses quantitative and qualitative data, it is necessary to indicate what type of Mixed Methods design is used.
Yours sincerely.
Author Response
Thank you for the good review of our manuscript!
We tried to answer all of your comments.
Comments and Suggestions for Authors
The work presented is interesting, however, it presents a series of difficulties that I believe do not make it suitable for publication
Answer: Thank you for your appreciation and comments to which we followed and we believe that we improved a lot.
- They only use physical variables. It has been shown that approximately 30% of sports performance is due to psychological variables. This is a fact that the authors acknowledge in the introduction.
Answer: Thank you for your comment, the reviewer is absolutely right. But in this article, the main emphasis was related to physical parameters, as we tried to validate the developed virtual environment for the purpose of diagnosing the skill level. First of all, we were interested in motor performance, reaction speed and puck kicking efficiency. As part of the study, we also conducted diagnostics of psychological qualities, in particular - the level of motivation among others. But it is the next stage research and we did not plan to review this data in this article since we had quite huge amount of information to analyse and report.
- The hypotheses are inadequately stated. Hypotheses should only be formulated on contrast statistics and qualitative analysis. Hypotheses should be formulated as follows: "Significant differences in [variable] will be determined between the control group and the experimental groups".
Answer: done. We reformulated 2-4 hypotheses as suggested by the reviewer.
- There are no references to the authorization of the ethics committee or to the Declaration of Helsinki.
Answer:
Indeed, we refer twice to the references to the authorization of the ethics committee as following ones:
- in the "Participants" section (p.5): «All of them participated on a voluntary basis with previously-signed consent, and with prior approval by the Ethical Committee of the Russian Psychological Society (2021/03), in accordance with the Helsinki Declaration.»
- Institutional Review Board Statement (p.15): The study was conducted in accordance with the Declaration of Helsinki and approved by the Ethics Committee and consent procedures of the Faculty of Psychology at Lomonosov Moscow State University (the approval No: 2021/03).
Additionally, we made a link to the document of the Declaration of Helsinki (p.5).
- The sample is very small.
Answer: We agree with the reviewer that the sample is small, and we write to Limitations about it. Nevertheless, the sample size was big enough to reach the statistical power of .80 for this study, as per our calculations.
Since the study involved hockey players with a high level of sportsmanship (M=14.2±2.3 years of average hockey experience, range: 8-20 years old), access to whom is rather limited – especially given the coronavirus pandemic – the sample was not large in total. Moreover, each test recording took a rather long period of time due to the need of putting on hockey equipment, fixing the sensors, and adjusting the VR equipment. It should also be noted that only two groups had an impact on the results. Some experts suggest that small sample size is not necessarily a major limitation as the available cohorts for both professional ice hockey players and professional wrestlers (and in particular their availability for research) have a ceiling due to expertise levels (Müller, Brenton & Rosalie (2015)).
- It is necessary to perform a generalizability analysis to determine the generalizability of the results.
Answer: We added the following text to the article:
«As part of this study, we obtained results reflecting the differences between professional ice hockey players and freestyle wrestlers in performing the task of hitting pucks at different level of difficulty. Since it was a pilot study, and because of the difficulty of accessing professional athletes, including COVID-19 limitations, the sample size was not huge, these results are difficult to generalize to a large sample.»
- Taking into account that it uses quantitative and qualitative data, it is necessary to indicate what type of Mixed Methods design is used.
Answer: The study used a mix methodology of quantitative and qualitative data analysis (we added to the text – p.8). This is due to the fact that in general there are not enough similar studies in the literature, and the availability of quantitative and qualitative data will allow us to study the problem raised in the study in more detail

Reviewer 3 Report
Dear authors,
I have completed the review of your manuscript. I found the article interesting, but think that several points can be improved to make it a substantial contribution to the field. In general, I appreciate the manuscript. The study brings additional knowledge to the field, but providing empirical support behind VR training among athletes.
______________
P1, L38-39: Authors need to explain more clearly the role of VR training and its associations with technical aspects of the sports. To me, authors skip to fast and take for granted such associations.
P2, L53-69: Need a stronger-better justification behind the relevance of RV training and the technical outcomes (e.g., kinematics). In other words, I think authors need to separate the mechanical outcomes (technique, hip movement, knee flexion) from perceptivo-cognitive aspects (e.g., reaction time, decision making, etc.) In summary, the rationale that lead to this study need to be improved, by giving more information and going further in the literature review.
Moreover, authors need to justify in a more appropriate way the choice of comparing wrestlers with hockey players (why not other sports, like martial arts or athletes who evolve in other team sport ?)...
...Also, some significant findings or past research could be cited:
Appelbaum, L.G., and Erickson, G. (2018). International Review of Sport and Exercise Psychology 11(1), 160-189.
Peric, T. (2020). Virtual reality as a mean of developing game performance in ice hockey, Research report.
Richard, V., Lavoie-Léonard, B., & Romeas, T. (2021). Embedding Perceptual–Cognitive Training in the Athlete Environment: An Interdisciplinary Case Study Among Elite Female Goalkeepers Preparing for Tokyo 2020. Case Studies in Sport and Exercise Psychology, 5(S1), S1-44.
Romeas, T., Guldner, A., & Faubert, J. (2016). 3D-Multiple Object Tracking training task improves passing decision-making accuracy in soccer players. Psychology of Sport and Exercise, 22, 1-9.
P2, L85-86: no studies focus on kinematic characteristics in VR settings. I agree with ice hockey, but was there other studies taken in other sports ?
P3, L90-104: As I mentioned earlier, some clarifications are needed in this paragraph, because this is the rationale of the study. Provide examples, of more details from the Tyreman study (or other researchers).
P3, L112: ...ice2 (remove 2 ath the end of the word)
P4, L139-140: Authors need to explain what blocks are. I understand that this seems the level of difficulty for VR exercicses, but this term appears suddenly. A more detailed explanation of the context need to be presented before this paragraph (or other terminology).
P4, L174-175: field hockey players need to be replaced by ice hockey players.
P4, L169: We used VR...with what kind of system ? Was it developed in a specific lab and/or was it previously validated ?
P5,L184-190: This information is too vague. What is is the specific task related with "hitting pucks" ? Is it like baseball ? If yes, then it does not seem specific to ice hockey, and I would recommend authors to be more specific here.
Methods: sample size is small. Did authors tests for a priori statistical power requirements ? If no (because this is a pilot study), then authors should support their choice.
Methods: Procedures are sufficiently explained, but nothing is presented about reliability of the VR exercises. Some improvements are possible here.
The results section is interesting, and provides insights that are in line with the study's objectives. We can really appreciate the differences between groups.
Table 1. Hitting pucks is not clear to me.
P12, L406-408: I think that the VR technology used in this study was not (hockey) specific enough. Some established, immersive VR technologies such as Sense Arena are promising and might be interesting for future research. I would like to see what authors consider about those kind of methodologies.
P13, L432: The "hitting" pucks task is still not clear to me. In this section, authors talk about shooting pucks (which is way more specific to hockey), but this kind of actions is not reported elsewhere. This can be improved.
Author Response
Thank you for the good review of our manuscript!
We tried to answer all of your comments.
Comments and Suggestions for Authors
Dear authors,
I have completed the review of your manuscript. I found the article interesting, but think that several points can be improved to make it a substantial contribution to the field. In general, I appreciate the manuscript. The study brings additional knowledge to the field, but providing empirical support behind VR training among athletes.
Answer: Thank you very much for the appreciation.
______________
P1, L38-39: Authors need to explain more clearly the role of VR training and its associations with technical aspects of the sports. To me, authors skip to fast and take for granted such associations.
Answer: Thank you for your comment. We have expanded and improved the introduction and added links to other studies.
P2, L53-69: Need a stronger-better justification behind the relevance of RV training and the technical outcomes (e.g., kinematics). In other words, I think authors need to separate the mechanical outcomes (technique, hip movement, knee flexion) from perceptivo-cognitive aspects (e.g., reaction time, decision making, etc.) In summary, the rationale that lead to this study need to be improved, by giving more information and going further in the literature review.
Answer:
Thank you very much for your comments. We corrected the introduction: we added references and separated the mechanical outcomes from perceptivo-cognitive aspects
Moreover, authors need to justify in a more appropriate way the choice of comparing wrestlers with hockey players (why not other sports, like martial arts or athletes who evolve in other team sport ?)...
Answer:
Analysing this topic, we found that most studies analyze professional athletes and novices in a particular sport. At the same time, there are very few studies that would show the presence of similar sports skills in different sports, but they do exist (we have added links to such studies in the text).
At the same time, analyzing the specific sports skills of hockey players (both motor and cognitive), it turned out that it is a common practice of professional hockey players to take training in martial arts, because they help them improve cognitive skills (anticipation, etc.)[1] [2] [3].
We had no specific criteria for choosing sports. We started with martial arts. Besides, we had access to these athletes. Given this, we decided at the beginning to compare the two sports and justify the transition from one sport to the other.
However, in the future we plan to include different sport types to compare.
...Also, some significant findings or past research could be cited:
Appelbaum, L.G., and Erickson, G. (2018). International Review of Sport and Exercise Psychology 11(1), 160-189.
Peric, T. (2020). Virtual reality as a mean of developing game performance in ice hockey, Research report.
Richard, V., Lavoie-Léonard, B., & Romeas, T. (2021). Embedding Perceptual–Cognitive Training in the Athlete Environment: An Interdisciplinary Case Study Among Elite Female Goalkeepers Preparing for Tokyo 2020. Case Studies in Sport and Exercise Psychology, 5(S1), S1-44.
Romeas, T., Guldner, A., & Faubert, J. (2016). 3D-Multiple Object Tracking training task improves passing decision-making accuracy in soccer players. Psychology of Sport and Exercise, 22, 1-9.
Answer: done. Thank you very much for the links. We have added links to them in the introduction.
P2, L85-86: no studies focus on kinematic characteristics in VR settings. I agree with ice hockey, but was there other studies taken in other sports?
Answer: Thank you for your comment. We have added links to research on kinematic characteristics in VR
P3, L90-104: As I mentioned earlier, some clarifications are needed in this paragraph, because this is the rationale of the study. Provide examples, of more details from the Tyreman study (or other researchers).
Answer: Done. We supplemented the text.
P3, L112: ...ice2 (remove 2 ath the end of the word)
Answer: done.
P4, L139-140: Authors need to explain what blocks are. I understand that this seems the level of difficulty for VR exercicses, but this term appears suddenly. A more detailed explanation of the context need to be presented before this paragraph (or other terminology).
Answer: we have added the information describing the blocks in the abstract
P4, L174-175: field hockey players need to be replaced by ice hockey players.
Answer: done. We have corrected the sentence: «The player's equipment corresponds to that of a team player (except goalkeeper), since the main goal of the study is to train and improve the individual motor skills of team players in ice hockey (except goalkeeper).».
P4, L169: We used VR...with what kind of system ? Was it developed in a specific lab and/or was it previously validated ?
Answer: done. We have corrected the sentence: «Virtual reality (VR) using SteamVR Tracking 2.0 system was used to study the differences between professional ice hockey players and novices in hockey with another sport background (freestyle wrestlers), in terms of motor responses on different levels of difficulty.»
This VR environment was developed in our laboratory as part of a research project [ПОЛИКАНОВА, И. С., ЛЕОНОВ, С. В., ЯКУШИНА, А. А., БУГРИЙ, Г. С., КРУЧИНИНА, А. П., ЧЕРТОПОЛОХОВ, В. А., & ЛЮЦКО, Л. Н. ВЕСТНИК МОСКОВСКОГО УНИВЕРСИТЕТА. СЕРИЯ 14: ПСИХОЛОГИЯ. ВЕСТНИК МОСКОВСКОГО УНИВЕРСИТЕТА, (1), 269-297]. http://msupsyj.ru/en/articles/article/9320/
P5,L184-190: This information is too vague. What is is the specific task related with "hitting pucks" ? Is it like baseball ? If yes, then it does not seem specific to ice hockey, and I would recommend authors to be more specific here.
Answer:
On the one hand the motor action is really similar to a hit in baseball. In hockey, however, many factors come into play when hitting the puck. For example, in hockey the puck can be sent from different sectors, different distances and different speeds (the hockey player does not know in advance where the puck will fly from). In hockey, the puck flies on the ice or low above the level of the ice. In our view, the cognitive task of hockey is very different from the cognitive task of baseball.
In our research we studied team players. Our goal was to identify a specific game situation and simulate it in a virtual environment. We chose the situation of a puck kicking on the goal line. This situation certainly does not cover the whole range of motor skills of a hockey player. But forwards and defensemen may be required to redirect pucks with their stick from time to time, in addition, field players may be on the goal line if the goalie is sent off the field.
Methods: sample size is small. Did authors tests for a priori statistical power requirements ? If no (because this is a pilot study), then authors should support their choice.
Answer:
The small sample size was due to a number of factors:
- Limited access to professional athletes, limited free time for professional athletes
- The limitations associated with COVID-19
- the duration of the experimental procedure (approximately 40 minutes per subject)
In the "Limitations" section, we refer to just some of the limitations of using VR: «Another important limitation of our study is the use of VR techniques to diagnose the professional skill of hockey players. VR is still a method that does not always show stable skill transfer results. Therefore, other indicators (cognitive tests, on-field exercises, etc.) should also be considered to assess the professional skill of hockey players. Further studies with increased sample size and adding other sports types, or people without any professional sport training, might help to better understand the differences in acquiring hockey professional skills by use of VR. Longitudinal studies would be beneficial to track the dynamics and speed of acquiring professional skills with use of VR.».
However, we checked and our sample size allowed to reach the coherent statistical power of 0.80.
Methods: Procedures are sufficiently explained, but nothing is presented about reliability of the VR exercises. Some improvements are possible here.
Answer: In order to develop the VR system, we were supported by professional hockey coaches who tested the system and suggested adjustments for VR exercises that were incorporated.
The results section is interesting, and provides insights that are in line with the study's objectives. We can really appreciate the differences between groups.
Table 1. Hitting pucks is not clear to me.
Answer: done. We have reformulated the title of the table
P12, L406-408: I think that the VR technology used in this study was not (hockey) specific enough. Some established, immersive VR technologies such as Sense Arena are promising and might be interesting for future research. I would like to see what authors consider about those kind of methodologies.
Answer: We are acquainted with Sensе Arena, but learned about it after we developed our own VR-PACE technology. We appreciate Sensе Arena's contribution to the training of hockey players. However, Sensе Arena, unlike our technology, does not record the hockey player's posture and detailed motor analysis.
P13, L432: The "hitting" pucks task is still not clear to me. In this section, authors talk about shooting pucks (which is way more specific to hockey), but this kind of actions is not reported elsewhere. This can be improved.
Answer: Thank you for your observation, it is corrected now. Sorry for the confuse. we corrected that linguistic mistake
[1] Martial Arts Instructor Jorge Blanco Teaches NHLers More Than Just How to Fight URL: https://www.si.com/nhl/2018/10/11/jorge-blanco-nhl-training-martial-arts (accessed on 20.11.2021).
[2] Martial Arts is changing the NHL URL: https://www.hockmansata.com/martial-arts-is-changing-the-nhl (accessed on 20.11.2021).
[3] The Science of the Skirmish: the Mechanics of a NHL Fight URL: https://www.ontheforecheck.com/2019/11/8/20951907/the-science-of-the-skirmish-the-mechanics-of-a-nhl-fight-mma-enforcers-strategy-ice-is-slippery (accessed on 20.11.2021).

Round 2
Reviewer 1 Report
I would like to thank the authors for introducing a number of changes to the manuscript indicated in the review. I believe that in its current version this work may be of interest to people who are looking for solutions supporting awareness and speed training. I have no more comments.
Reviewer 2 Report
Thank you for your efforts. However, you should make the following changes:
1. Upload the latest version of article. It is now as supplementary material.
2. The sample is very small and that is an important limitation. In addition to putting it as a limitation, they should perform a generalizability analysis to justify that the results are generalizable.
3. They should rephrase the hypotheses, in the following sense "The existence of significant differences in [variable] between the control group and the experimental groups will be determined."
Sincerely.